# Design, Synthesis, Molecular Modeling and Anti-Hyperglycemic Evaluation of Quinazoline-Sulfonylurea Hybrids as Peroxisome Proliferator-Activated Receptor Gamma (PPARγ) and Sulfonylurea Receptor (SUR) Agonists

**DOI:** 10.3390/ijms23179605

**Published:** 2022-08-24

**Authors:** Mohamed Ayman El-Zahabi, Faida H. Bamanie, Salah Ghareeb, Heba K. Alshaeri, Moudi M. Alasmari, Mohamed Moustafa, Zohair Al-Marzooki, Mohamed F. Zayed

**Affiliations:** 1Pharmaceutical Chemistry Department, Faculty of Pharmacy, Al-Azhar University, Cairo 11884, Egypt; 2Biochemistry Department, College of Medicine, King Abdulaziz University, Jeddah 21589, Saudi Arabia; 3Pharmacology Department, Faculty of Pharmacy, Zagazig University, Zagazig 44519, Egypt; 4Pharmaceutical Sciences Department, Fakeeh College for Medical Sciences, Jeddah 21461, Saudi Arabia; 5College of Medicine, King Saud Bin Abdulaziz University for Health Sciences (KSAU-HS), Jeddah 21461, Saudi Arabia; 6King Abdullah International Medical Research Center (KAIMRC), Jeddah 21423, Saudi Arabia; 7Medicinal Chemistry Department, Faculty of Pharmacy, Mansurah University, El Mansurah 35516, Egypt

**Keywords:** design, synthesis, quinazoline, sulfonylurea, anti-hyperglycemic

## Abstract

New quinazoline-sulfonylurea hybrids were prepared and examined for their in vivo anti-hyperglycemic activities in STZ-induced hyperglycemic rats using glibenclamide as a reference drug. Compounds VI-6-a, V, IV-4, VI-4-c, IV-6, VI-2-a, IV-1, and IV-2 were more potent than the reference glibenclamide. They induced significant reduction in the blood glucose levels of diabetic rats: 78.2, 73.9, 71.4, 67.3, 62, 60.7, 58.4, and 55.9%, respectively, while the reference glibenclamide had 55.4%. Compounds IV-1, VI-2-a, IV-2, V, and IV-6 showed more prolonged antidiabetic activity than glibenclamide. Moreover, molecular docking and pharmacokinetic studies were performed to examine binding modes of the prepared compounds against peroxisome proliferator-activated receptor gamma (PPARγ). The highest active compounds exhibited good binding affinity with high free energy of binding against PPARγ. In silico absorption, distribution, metabolism, elimination and toxicity (ADMET) studies were performed to investigate pharmacokinetics and safety of the synthesized compounds. They showed considerable human intestinal absorption with low toxicity profile.

## 1. Introduction

Type 2 diabetes is a chronic metabolic disorder in which there is a high level of glucose in the body due to impaired insulin secretion. It affects millions of people worldwide, and it is the most common type of diabetes [1,2,3,4,5]. Type 2 diabetes is more common in older adults, but also affects kids and teens due to childhood obesity [6,7]. Controlling diet and practicing exercise along with oral hypoglycemics are first line of treatment of type 2 diabetes to adjust blood glucose levels and avoid diabetic health complications [8,9,10,11]. There are many classes of oral hypoglycemic agents, e.g., sulfonylurea, biguanides, meglitinides, thiazolidinediones, dipeptidyl peptidase-4 (DPP-4) inhibitors, glucagon-like peptide-1 (GLP-1) receptor agonists, and sodium-glucose co-transporter-2 (SGLT2) inhibitors [11,12,13,14]. Sulfonylurea is a well-known mainstay therapy of type 2 diabetes. The second-generation sulphonyl urea glibenclamide is widely used worldwide by stimulating insulin secretion from β-cells of the pancreas [9,10]. It binds with sulfonylurea receptors (SUR) and ATP-sensitive K+ (KATP) channels in pancreatic β-cells to stimulate insulin secretion [11]. Hence, medications which stimulate insulin secretion boosted by high glucose level with low side effects will be highly valued [11,12]. Furthermore, Sulfonylurea derivatives produce their biological activity through binding with peroxisome proliferator-activated receptors gamma (PPARγ) in adipocytes [12,13]. Glibenclamide and other derivatives were reported as PPARγ receptors agonists [10,11,12,13,14,15,16]. A pharmacophoric model of these derivatives was designed to include three lipophilic parts separated by anionic and amidic linkers [13,14]. Figure 1 shows the main structural features of these derivatives. PPARγ is a well-recognized member of the PPAR subfamily. It has an important role in the treatment of diabetes mellites by increasing insulin sensitivity in different tissues. The pharmacophoric characters of PPARγ agonist include an acidic head joined with small linker to a heteroaromatic lipophilic tail [15,16,17]. Thiazolidinedione derivatives as rosiglitazone and pioglitazone are reported as PPARγ agonists [18,19,20,21,22,23]. Moreover, many quinazoline derivatives have been reported to possess a marked hypoglycemic action [24,25,26,27,28]. Centpiperalone [29,30], and Zenarestate [30] were tried in clinical traials as antidiabetic agents. Bunazosin, another quinazoline derivative, which is principally used as an alpha blocker, was found to possesses beneficial effects on lipid and glucose metabolism [30].

In addition, it was reported that the hybrid structures of 2-substituetd 4(3H)-quinazolinone and sulfonylurea can serve as effective hypoglycemic compounds [20]. Some of the recent compounds which showed good antidiabetic activity are the 3-[4-[[[cyclohexylamino)carbonyl]-aminosulfonyl]phenyl]-4-(3H)-quinazolinones] and other heterocyclic derivatives [31,32,33,34]. Figure 2 show chemical structures and structural features of Glibenclamide and Rosiglitazone [34,35]. Taken together, this study reports synthesis of new quinazoline-sulfonylurea hybrids with the aim of discovering new agents having strong agonistic activity against PPARγ receptors, suitable pharmacokinetic properties, and low toxicity profile.

### Rationale

A previous work (Figure 3) reported effective hypoglycemic agents formed of hybrid structures of quinoxaline and quinazoline nuclei joined with sulphonyurea derivatives [34,35]. Based on this work, and ligand-based drug design approach, we designed novel hypoglycemic agents having the following structural features:Quinazolinone nucleus as a hydrophobic domainAromatic sulfonylurea or sulphonamide group miming sulfonylurea derivativesDifferent attached hydrophobic groups with different electronic environment to study and compare the difference in their biological activities.

There are a lot of structural similarities between newly designed compounds and the well-known hypoglycemic agents glibenclamide and rosiglitazone to get a hybrid molecule collecting the required features to work effectively as SUR and PPARγ agonists. Figure 4 explains structural similarities between newly synthesized compounds, and reported hypoglycemics glibenclamide and rosiglitazone. It shows that the acidic tail of rosiglitazone was replaced by acidic moiety of sulfonylurea. It is essential for activity, because it forms hydrogen bonds with Tyr473, Tyr327 and His323. Aromatic and lipophilic groups in SUR agonist are replaced by different lipophilic groups to maintain the required hydrophobic interactions. In addition, the NH group in sulphamoyl moiety has acidic PKa (4.9–6.5) and this leads to ionization at physiological PH (7.4). The ionization process produces an ionic linker helps in SUR agonistic activity [14,15]. The SO_2_ group acts as a spacer between the acidic NH group and the aromatic group which gives the optimal length for PPARγ agonistic activity [16,17]. A disubstituted aromatic ring at para position (1,4) is required for optimum binding with PPARγ [36,37], and has an essential role for binding with SUR receptors. Two groups were designed to contain a lipophilic linker (-CH_2_-CH_2_-) between the aromatic scaffold and the lipophilic tail, while another group contains no linker, but an aromatic scaffold joined directly with lipophilic tail to investigate structural activity relationships (SAR). The quinazoline nucleus was used as a lipophilic center needed for agonistic activity with PPARγ and SUR. R1, R2 are different lipophilic groups with different electronic environment to compare their activity and study SAR. Alignment of newly synthesized compounds (VI-3b) with glibenclamide showed excellent matching features Figure 5.

This study focused on the discovery of new hypoglycemic agents having structural similarities with both of SUR and PPARγ agonists to get the benefits of a dual acting mechanism aiming of finding safe and effective hypoglycemic agents. The general structure of these compounds includes a bulky heterocyclic system, namely quinazolinone, joined with sulfonylurea or sulphonamides group via an aromatic spacer and a lipophilic linker to explore the hypoglycemic activity.

## 2. Results and Discussion

### 2.1. Chemistry

The synthesis of target compounds is depicted in Figure 1, Figure 2, Figure 3 and Figure 4. Many 4H-3,1-benzoxazin-4-ones (II) were obtained according to Figure 1 by reacting anthranilic acid with two moles of benzoyl chloride pyridine (method A) [23,24,25], or by refluxing anthranilic acid with certain acid anhydrides namely, acetic anhydrides (method B) [27], or by cyclization of N-acylanthranilic acids (I) with acetic anhydrides (method C) [24].

Trials to condense the bezoxazinone II-a with 4-(2-aminoethyl)benzenesulfonamide in order to get the target intermediate 2-Phenyl-3-(p-aminosulfonyl]phenylethyl)-2-phenyl-4(3H)-quinazolinone were summarized in Figure 2 as follows:

Figure 2 explain four methods for the preparation of target compounds:(1)Overnight reflux in methanol or ethanol or DMF afforded the corresponding open structure 2-benzamido-N-(p-aminosulfonyl]phenylethyl)-benzamide III-1 with trace amount of the corresponding cyclic structure IV-1.(2)Refluxing in dry pyridine for 4 h afforded a mixture of both the open and cyclic intermediates III-1 and IV-1, respectively, where the cyclic intermediate was the major.(3)Overnight reflux in glacial acetic acid afforded the cyclic IV-1 as a major product.(4)Fusion for two hours at appropriate temperature gave the target cyclic intermediate IV-1 in best yields in comparison with other cyclization procedures.

Fusing the quinazolinone IV-a with 4-(2-aminoethyl)benzenesulfonamide gave IV-1 in about 70%. Using acetic acid as the refluxing solvent gave the product in only 45%. While using pyridine, the product was synthesized in about 40%. Other refluxing solvents such as methanol, ethanol and DMF gave the open structure III-1 as a major product. Using toluene or xylene was not a good idea because the starting materials were not soluble in both solvents and the reaction gave a sticky mass. TlC of the sticky mass showed a messy feature including the starting material. Parallel fusion of the benzoxazine II-a with sulfanilamide resulted in product VI-1 in almost parallel results to that conducted with 4-(2-aminoethyl)benzenesulfonamide. 3-(p-Sulfamoylaralkyl)-4-(3H)-quinazolinones IV-(1-6) were prepared by nucleophilic displacement reaction through fusion of homosulfanilamide derivatives or sulfanilamide derivatives with the benzoxazinones II-(1–6) to obtain the target compounds as shown in Figure 3. Sulfonyl urea derivatives of 3-(p-Sulfamoylaralkyl)-4-(3H)-quinazolinones (VI-1 to 6) were prepared according to reported procedures [25] as explained in Figure 4.

### 2.2. Biology

#### 2.2.1. In Vivo Antihyperglycemic Screening

The antidiabetic activity of the synthetized compounds of quinazoline nucleus from III-1 to VII was determined in diabetic rats according to the method of Villar et al., (1986) [36,37,38,39]. Glibenclamide was used as a positive control to measure the antidiabetic activity of the target compounds. The experiment was performed on two levels. The first level was performed to study the effect of a small dose, 2 mg/kg. The second level was performed to study the effect of discontinuation of drugs for 6 days. The first level indicates the potency of tested compounds in comparison with the reference glibenclamide, while the second level indicates the prolonged action of the newly synthesized compounds versus the reference glibenclamide.

##### Effect of Glibenclamide and the Tested Compounds in a Daily Dose of 2 mg/kg for 6 Days on Blood Glucose Levels of Diabetic Rats

Results in table one after using STZ in a dose of 55 mg/kg. This dose induced significant elevation in glucose blood levels of rats and induced diabetes recording at 406 to 475 mg/dL by the end of the experiment. Glibenclamide significantly decreased the blood glucose level by 55.4% of the value before treatment. Some of the tested compounds induced significant reduction in the blood glucose levels of diabetic rats. This effect ranged from 78.2 to 5.8% from the values before treatment. These compounds are III-1, IV-1, VI-1-a, IV-2, VI-2-a, VI-2-b, IV-3, VI-3-b, VI-3-c, IV-4, VI-4-b, VI-4-c, IV-5, VI-5-a, VI-5-b, IV-6, VI-6-a, VI-6-b, V, VII. These compounds can be arranged in a descending order according to their relative potency (RP) to that of glibbnclamide as follows: VI-6-a > V > IV-4 > VI-4-c > IV-6 > VI-2-a < IV-1 > IV-2 > glibenclamide > IV-3 > VI-2-b > IV-5 > VI-5-a > VII > VI-3-b > VI-4-b > VI-6-b > VI-1-a < VI-3-c > VI-5-b > III-1. Compounds No VI-6-a, V, IV-4, VI-4-c, IV-6, VI-2-a, IV-1, IV-2 were more potent than that of glibenclamide.

It is apparent that compounds VI-6-a, V, IV-4, VI-4-c, IV-6, VI-2-a, IV-1 and IV-2 are very promising compounds as they are more potent than glibenclamide by the end of the sixth day of treatment at 2 mg/kg dose level in diabetic rats. The maximum glucose lowering percentage by the end of the sixth day is 78.2% in comparison with a 55.4% decrease for glibenclamide. Figure 6 shows relative potency of the synthesized compounds in comparison to the reference glibenclamide.

##### Effect of Discontinuation of Drugs for 6 Days on the Rebound Elevation of Blood Glucose of Diabetic Rats

The discontinuation of glibenclamide induced slight non-significant rebound elevation of the blood glucose to 183 mg/dL vs. 150 mg/dL before the discontinuation, but the drug still had significant antidiabetic activity (Table 1). The tested compounds have variable effects concerning this aspect of drug withdrawal and rebound elevation of blood glucose of rats. Compounds V, VII, IV-2, VI-2-a, IV-3, IV-6, IV-1 and VI-1-a still have blood glucose lowering activity after their long-term suspending of their administration. The blood glucose of the treated groups was slightly non-significantly elevated but still below the values before treatment. In respect of these effects, these compounds can be arranged in a descending order according to their relative potency to glibenclamide as follows: IV-1> VI-2-a > IV-2 > V > IV-6 > glibenclamide > IV-3. The prolonged effect of these compounds may explain the low rate of elimination and/or increased their plasma protein binding affinity and maybe their half-lives. On the other hand, the antidiabetic activity of some of the tested compounds was partially of totally abolished and the blood glucose levels of the treated animals returned to their levels before treatment or maybe slightly elevated. These compounds are VI-2-b, VI-3-b, IV-4, VI-4-b, VI-4-c, IV-5, VI-5-b, VI-6-a, VI-6-b and III-1. Moreover, in rats treated with compounds, VI-4-b, IV-5, III-1 became severely diabetic after discontinuation of the treatment. The surprising effect of some of these compounds (IV-4, VI-4-c, VI-6-a and VI-6-b) is that they have rapid onset and intensive antihyperglycemic effect but with short duration of action.

Figure 7 show that compounds IV-1, VI-2-a, IV-2, V, and IV-6, respectively, have maintained their blood glucose-lowering effects after 6 days of discontinuation. This means that substitution with phenyl, methyl and benzyl groups at the 2-position of the quinazolinone nucleus maintained their activity. The prolonged effect of these compounds may be explained in terms of the low rate of elimination and/or increased plasma protein binding affinity and maybe their half-lives.

It is of interest to note that compound IV-4 and IV-5 or n-propyl and isopropyl substituents at the 2-position of the quinazolinone ring did not maintain its blood lowering effect after discontinuation. This may be attributed to the rapid biotransformation and elimination of these compounds.

### 2.3. Molecular Modeling

Based on the availability of PPARγ crystal structures in the Protein Data Bank, receptor-based drug design (docking) technique [37,38,39] was used to explore the binding mode of the synthesized compounds with the potential target, PPARγ.

#### 2.3.1. Modeling Studies

The crystal structure of PPARγ was retrieved from Protein Data Bank [PDB ID- 1FM6, resolution 2.1 Å] (http://www.pdb.org, 10 May 2019), and considered as a target for docking simulations. The docking analysis was performed using MOE [40] software to evaluate the free energies and binding mode of the designed molecules against PPARγ. At first, the crystal structure of PPARγ was prepared by removing water molecules and retaining only one chain and its co-crystallized ligand, rosiglitasone. Then, the protein structure was protonated, and the hydrogen atoms were hidden. Next, the energy was minimized, and the binding pocket of the protein was defined. The 2D structures of the synthesized compounds and reference ligands (rosiglitazone and glibenclamide) were sketched using ChemBioDraw Ultra 14.0 and saved as MDL-SD format. Then, the saved file was opened using MOE and 3D structures were protonated. Next, energy minimization was applied. Before docking the synthesized compounds, validation of the docking protocol was carried out by running the simulation only using the co-crystallized ligand and low RMSD between docked and crystal conformations. The molecular docking of the synthesized compounds and the cocrystallized ligand was performed using a default protocol. In each case, 10 docked structures were generated using genetic algorithm searches. The PPARγ cavity consists of three main parts; an entrance, arm I and arm II. The arm I contains four polar residues, Tyr473, Ser289, His449 and His323, involved in hydrogen bonding. Arm II comprises Leu353, Ile 341, Ile281 and Val339 while the entrance consists of Arg288, Ser342, Leu330and Leu333 [38,39]. The results of docking studies revealed that the synthesized compounds exhibited similar orientations inside the putative binding sites of PPARγ. The designed members showed good binding energies ranging from −12.48 to −24.81 kcal/mol (Table 2). The proposed binding mode of the co-crystallized ligand, rosiglitazone, showed an affinity value of −26.29 kcal/mol. The thiazolidinedione moiety was oriented in the polar arm I of the target receptor. Hydrogen bonding interaction with a distance of 1.74 Å and scoring of 55% was observed between the acidic hydrogen of the imide group and Tyr473. The oxygen atom of the first carbonyl group in thiazolidinone moiety formed one hydrogen bond with His449 with a distance of 2.78 Å and scoring 11%. The oxygen atom of the second carbonyl group in the thiazolidinone moiety formed a hydrogen bond with His323 with a distance of 2.35 Å and scoring 27%. The pyridine moiety occupied the area of the hydrophobic pocket. These results agreed with the reported data [38,39] (Figure 8). The mapping surface technique was carried out to show rosiglitazone occupying the active pocket of PPARγ (Figure 9).

Compound IV-5 as representative example exhibited a binding mode similar to that of rosiglitazone, with an extra hydrogen bond and affinity value of −12.48 kcal/mol. The sulfonamide group was oriented in the polar arm I of PPARγ and formed four hydrogen bonds, two with Tyr473 via SO_2_ group (distances 2.93 Å, 2.86 Å, scores = 15%, 14%), a hydrogen bond with Tyr327 via the NH group (distance = 1.51, score = 72%), and a hydrogen bond with His323 via SO_2_ (distance = 2.49, score = 29%). The quinazolinone moiety was oriented in the hydrophobic entrance forming pi-pi and pi-cation interactions with Arg288 (Figure 10). Figure 11 shows the 3D shape of compound IV-5 occupying the active pocket of PPARγ.

#### 2.3.2. In Silico ADMET Studies

##### Evaluation of Lipophilicity, Solubility, Drug Likeness, GIT Absorption and Toxicity Risk

The SwissADME [41] is a software package used for the prediction of different pharmacokinetic parameters, such as as lipophilicity, solubility, drug likeness, and bioavailability. The results of these parameters are shown in Table 3. Results indicated that most of the newly synthesized compounds have good lipophilic characters and follow the Lipiniski rule. Additionally, T.E.S.T [42] is another software package used for the prediction of the toxicity profile of the chemical compounds. All the compounds had a negative toxicity risk based on the results obtained from this analysis as shown in Table 3.

### 2.4. Structural Activity Relationship

In group IV, the propyl analog (IV-4) was the most potent among all six derivatives of the quinazolinone ring analogs followed by the benzyl (IV-6) and then the methyl. The least active derivative among all was the isopropyl derivative (IV-5).Comparing compound (IV-1) (in which two methylene spacers are present between the quinazolinone ring and the phenyl sulfonamide group) and compound V which no spacers has unexpectedly shown that the two ethylene bridge was not essential for activity.There was big difference between the open structure (III-1) and its cyclic congener (IV-1), which showed more than ten folds the activity of the corresponding open structure.Compounds having benzyl, phenyl, n-propyl and methyl groups at position 2 of quinazolinone nucleus were the most potent among all groups.These compounds are devoid of the important chloro group which is essential for our design, activity and duration of action as reported [24,25,26].Compounds (IV-1), (VI-2-a), (IV-2), (V), (IV-6) which have phenyl, methyl and benzyl groups at the 2-position of the quinazolinone nucleus have shown prolonged actions while compounds (IV-4) and (IV-5) (having the n-propyl or isopropyl groups at 2-position of the quinazolinone nucleus) showed short duration of action.These compounds are so promising that this work could be performed by modifying the 5-position of the quinazolinone nucleus and attaching other groups to the sulfonamide group.

## 3. Materials and Methods

### 3.1. Chemistry

Proton Nuclear Magnetic Resonance spectra (^1^HNMR) were recorded using Me_4_Si as internal standard at Faculty of Science, King Abdul Aziz University and some of the NMR data were carried out in Faculty of Pharmacy, King Saud University. Melting points were determined with Barnested electrothermal melting point apparatus and are uncorrected. Elemental Analysis was performed by microanalytical laboratory at faculty of Science, King AbdulAziz University. The chemicals, reagents and solvents were purchased from Aldrich chemical company and other international companies through Al-Thwabit and Bayouni/trading companies in Saudi Arabia.

#### 3.1.1. *N*-Acylnthranilic Acids (I)

The method of Steiger [27] was used for the preparation of *N*-propionyl, *N*-phenacetyl, *N*-butyryl, *N*-isobutyrylanthranilic acids.

#### 3.1.2. 4H-3,1-Benzoxazin-4-Ones (II)

The appropriate N-acylanthranilic acid (II) (0.05 mole) was refluxed with acetic anhydride for one hour according to reported procedures [24,28]. The reaction mixture was distilled to about half its volume. On cooling, the corresponding benzoxazinones **II** were afforded.

#### 3.1.3. *o*-Benzamido-*N*-(*p*-aminosulfonyl]phenylethyl)-benzamide (III-1)

A mixture of equimolar quantities of the appropriate benzoxazinone (II) and the corresponding sulfanilamide or 4-(2-Aminoethyl)benzenesulfonamide (0.01 mole) were refluxed with EtOH. The starting materials initially dissolved in the alcohol then precipitate after about one and a half hours. The mixture was refluxed overnight, and the solvent was concentrated, filtered and washed with hot alcohol. Compound III-1 was obtained as a pure solid in high yield. Recrystallization for further purification from MeOH afforded colorless needle crystals, m.p. 253–255° in over 80% yield.

^1^H NMR of (III-a): DMSO-d_6_, d 2.94–2.98 (t, 2H, CH_2_), 3.49–3.58 (q, 2H, CH_2_), 7.18–7.22 (t, 2H, Arom-H), 7.27–7.39 (brs, 1H, SO_2_NH_2_), 7.44–7.46 (d, 2H, j = 9.5 Hz, Arom-H), 7.55–7.66 (d, 4H, Arom-H), 7.74–7.77 (t, 2H, Arom-H), 7.92–7.94 (d, 2H, j = 9.5 Hz, Arom-H), 8.62–8.64 (d, 2H, j = 10. Hz, Arom-H), 9.00 (br, t-like, 1H, NH-CH_2_), 12.45 (brs, 1H, NH-CO-Ph). C_21_H_21_N_3_O_4_S (423.48).

#### 3.1.4. General Procedures for Synthesis of 3-(*p*-Sulfamoylaralkyl)-4-(3H)-quinazolinones (IV-1–6)

A mixture of equimolar quantities of the appropriate benzoxazinone (II) and the corresponding sulfanilamide or 4-(2-Aminoethyl)benzenesulfonamide (0.01 mole) were fused in a two neck flask attached to a reflux condenser for 1–3 h. The solid obtained was crystallized from ethanol to give the corresponding quinazolinones (**IV-1–6**).

##### 2-Phenyl-3-(*p*-aminosulfonyl]phenylethyl)-2-phenyl-4(3H)-quinazo-linone (IV-1)

White crystals (yield 70%); m.p. 143–145 °C; 1H NMR (DMSO-d6, 600 MHz) δ (ppm): 2.92 (br, 2H, CH_2_), 4.10 (br, 2H, CH_2_), 6.98 (br, 2H, NH_2_) 7.32 (br, 2H, Arom-H), 7.46–7.63 (br, m, 9H, Arom-H), 7.85 (br, 1H, Arom-H), 8.23 (br, 1H, Arom-H); Anal. Calcd. for C_22_H_19_N_3_O_3_S (405.47): C, 65.17; H, 4.72; N, 10.36; Found: C, 65.47; H, 5.16; N, 9.67.

##### 2-Methyl-3-(*p*-aminosulfonyl]phenylethyl)-4(3H)-quinazolinone (IV-2)

White crystals (yield 72%); m.p. 258–260 °C; ^1^H NMR (DMSO-d_6_, 600 MHz) δ (ppm): 2.52 (s, 3H, CH_3_), 3.00-3.10 (br, t-like, 2H, CH_2_), 4.2–4.3 (br, t-like, 2H, CH_2_), 7.34–7.41 (brs, 2H, NH_2_), 7.56–7.58 (d, 1H, Arom-H), 7.75–7.82 (m, 3H, Arom-H), 8.11–8.12 (d, 1H, Arom-H); Anal. Calcd. for C_17_H_17_N_3_O_3_S (343.4): C, 59.46; H, 4.99; N, 12.24; Found: C, 59.34; H, 5.21; N, 12.39.

##### 2-Ethyl-3-(*p*-aminosulfonyl]phenylethyl)-4(3H)-quinazolinone (IV-3)

White crystals (yield 68%); m.p. 256–258 °C; ^1^H NMR (DMSO-d_6_, 600 MHz) δ (ppm): 2.52 (s, 3H, CH_3_), 3.00–3.10 (br, t-like, 2H, CH_2_), 4.2–4.3 (br, t-like, 2H, CH_2_), 7.34–7.41 (brs, 2H, NH_2_), 7.56–7.58 (d, 1H, Arom-H), 7.75–7.82 (m, 3H, Arom-H), 8.11–8.12 (d, 1H, Arom-H); Anal. Calcd. for C_18_H_19_N_3_O_3_S (357.43): C, 60.49; H, 5.36; N, 11.76; Found: C, 60.78; H, 5.06; N, 11.57.

##### 2-n-Propyl-3-(*p*-aminosulfonyl]phenylethyl)-4(3H)-quinazolinone (IV-4)

White crystals (yield 65%); m.p. 224–226 °C; ^1^H NMR (DMSO-d_6_, 600 MHz) δ (ppm): 0.96–0.99 (t, 3H, CH_3_), 1.75–1.79 (Sixtet-like, 2H, CH_2_), 2.73–2.76 (t, 2H, CH_2_), 3.04–3.07 (t, 2H, CH_2_), 4.25–4.28 (t, 2H, CH_2_), 7.33 (brs, 2H, NH_2_), 7.47–7.49 (d, 2H, Arom-H), 7.50–7.51 (d, 2H, Arom-H), 7.60–7.61 (d, 2H, Arom-H), 7.77–7.79 (d, 3H, Arom-H), 8.13–8.14 (d, 1H, Arom-H); Anal. Calcd. for C_19_H_21_N_3_O_3_S (371.45): C, 63.40; H, 5.73; N, 11.38; Found: C, 63.43; H, 6.22; N, 11.52.

##### 2-*iso*-Propyl-3-(*p*-aminosulfonyl]phenylethyl)-4(3H)-quinazolinone (IV-5)

White crystals (yield 70%); m.p. over 300 °C; ^1^H NMR (DMSO-d_6_, 600 MHz) δ (ppm): 1.18–1.19 (d, 6H, CH_3_), 3.04–3.07 (m, 3H, CH, CH_2_), 7.33 (brs, 2H, NH_2_), 7.44–7.46 (d, 2H, Arom-H), 7.48–7.51 (t, 1H, Arom-H), 7.60–7.61 (d, 1H, Arom-H), 7.76–7.80 (m, 3H, Arom-H), 8.14–8.15 (d, 1H, Arom-H).; Anal. Calcd. for C_19_H_21_N_3_O_3_S (371.45): C, 63.40; H, 5.73; N, 11.381; Found: C, 63.43; H, 6.22; N, 11.52.

##### 2-*iso*-Propyl-3-(*p*-aminosulfonyl]phenylethyl)-4(3H)-quinazolinone (IV-6)

White crystals (yield 65%); m.p. 143–145 °C; ^1^H NMR (DMSO-d_6_, 600 MHz) δ (ppm): 2.77–2.80 (t, 2H, CH_2_), 4.14–4.17 (t, 2H, CH_2_), 4.24 (s, 2H, CH_2_), 7.26–7.36 (m, 9H, Arom-H+ NH_2_), 7.43–7.46 (t, 1H, Arom-H), 7.51–7.54 (d, 1H, Arom-H), 7.81–7.82 (d, 3H, Arom-H), 8.16-8.18 (d, 1H, Arom-H); Anal. Calcd. for C_23_H_21_N_3_O_3_S (419.5): C, 63.40; H, 5.73, 11.38; Found: C, 63.43; H, 6.22; N, 11.52.

##### 2-Phenyl-3-(*p*-sulfonyl]phenyl)-4(3H)-quinazolinone (V)

White crystals (yield 65%); m.p. 250–252 °C; ^1^H NMR (DMSO-d_6_, 600 MHz) δ (ppm): 7.26–7.32 (m, 3H, Arom-H), 7.41–7.43 (d, 2H, Arom-H), 7.46 (brs, 2H, NH_2_), 7.57–7.59 (d, 2H, Arom-H), 7.61–7.64 (t, 1H, Arom-H), 7.75–7.77 (d, 2H, Arom-H), 7.79–7.81 (d, 1H, Arom-H), 7.91–7.94 (t, 1H, Arom-H), 8.21–8.23 (d, 1H, Arom-H); Anal. Calcd. for C_20_H_15_N_3_O_3_S (377.42): C, 63.65; H, 4.01, 11.13; Found: C, 63.32; H, 3.86; N, 11.00.

#### 3.1.5. General Procedures for Synthesis of Sulfonyl Urea Derivatives of 3-(*p*-Sulfamoylaralkyl)-4-(3H)-quinazolinones (VI-1–6)

A mixture of (IV-1-6) (0.05 mole) and anhydrous K_2_CO_3_ (0.1 mol, 13.8 g) in dry acetone (100 mL) was stirred and refluxed for 2 h. At this temperature, a solution of alkyl or aryl isocyanate (0.075 mole) in dry acetone (20 mL) was added in a dropwise manner. The reaction mixture was further refluxed overnight. The solvent was then removed under vacuum and the solid residue was dissolved in water and then acidified with a sufficient amount of 2N HCl. The separated product was recrystallized from ethanol.

##### 2-Phenyl-3-[4-[[[(phenylamino)carbonyl]aminosulfonyl]phenylethyl]-4(3H)-quinazolinone (VI-1-a)

White crystals (yield 68%); m.p. 136–138 °C; ^1^H NMR (DMSO-d_6_, 600 MHz) δ (ppm): 3.27–3.34 (t-like, 2H, CH_2_), 4.16 (br, 2H, CH_2_), 6.36 (br, 1H, NH) 7.27–734 (br, 10H, Arom-H), 7.46–7.63 (br, m, 9H, Arom-H), 7.52–7.64 (d-like, 3H, Arom-H), 7.83 (br, 4H, Arom-H), 8.15 (s, 1H, Arom-H), 10.36 (brs, 1H, NH); Anal. Calcd. for C_29_H_24_N_4_O_4_S (524.59): C, 66.40; H, 4.61, 10.68; Found: C, 65.92; H, 4.46; N, 10.32.

##### 2-Phenyl-3-[4-[[[(cyclohexylamino)carbonyl]aminosulfonyl]phenyl]-4(3H)-quinazolinone (VI-1-b)

White crystals (yield 62%); m.p. 165–167 °C; ^1^H NMR (DMSO-d_6_, 600 MHz) δ (ppm): 0.95–1.25 (m, 6H, 3CH_2_), 11.42–1.70 (m, 4H, 2CH_2_), 2.92–3.0 (br, 2H, CH_2_), 3.20–3.35 (br, 1H, N-CH), 4.11 (br, 2H, CH_2_), 6.42–6.55 (br, d like, NH-CH), 6.93–7.05 (br, d-like, 2H, Arom-H), 7.29–7.38 (br, m, 2H, Arom-H), 7.48–7.55 (br, m, 2H, Arom-H), 7.57–7.62 (br, q-like, 1H, Arom-H),7.63–71(br, m, 3H, Arom-H), 7.82–7.89 (br, t-like, 1H, Arom-H), 8.20–8.24 (br, d, 1H, Arom-H), 10.2–10.5 (br, 1H, NH); Anal. Calcd. for C_29_H_30_N_4_O_4_S (530.64): C, 65.64; H, 5.70, 10.56; Found: C, 65.52; H, 5.95; N, 10.19.

##### 2-Phenyl-3-[4-[[[(*n*-butylamino)carbonyl]aminosulfonyl]phenylethyl]-4(3H)-quinazolinone (VI-1-c)

White crystals (yield 70%); m.p. 163–165 °C; ^1^H NMR (DMSO-d_6_, 600 MHz) δ (ppm): 0.77–0.80 (t, 3H, CH_3_), 1.1–1.2 (m, 2H, CH_2_), 1.25–1.32 (m, 2H, CH_2_), 2.86–2.95 (br, 2H, CH_2_), 3.20–3.35 (br, 2H, N-CH_2_), 4.12 (br, 2H, CH_2_), 6.40 (br, NH-CH_2_), 7.0–7.13 (br, d, 2H, Arom-H), 7.35–7.40 (br, d, 2H, Arom-H), 7.45–7.50 (br, t, 3H, Arom-H), 7.52–7.75 (m, 4H, Arom-H), 7.83–8.90 (t, 1H, Arom-H), 8.20–8.25 (d, 1H, Arom-H), 10.2–10.7 (br, 1H, NH); Anal. Calcd. for C_27_H_28_N_4_O_4_S (504.60): C, 64.27; H, 5.59; N, 11.10; Found: C, 64.45; H, 6.30; N, 10.41.

##### 2-Methyl-3-[4-[[[(phenylamino)carbonyl]aminosulfonyl]phenylethyl]-4(3H)-quinazolinone (VI-2-a)

White crystals (yield 66%); m.p. 136–138 °C; ^1^H NMR (DMSO-d_6_, 600 MHz) δ (ppm): 2.63 (s, 3H, CH_3_), 3.05–3.09 (t, 2H, CH_2_), 4.25–4.29 (t, 2H, CH_2_), 6.94–6.98 (t,2H, Arom-H), 7.25–7.29 (t, 2H, Arom-H), 7.44–7.49 (d, 2H, Arom-H), 7.51–7.57 (t, 3H, Arom-H), 7.65–7.67 (d, 1H, Arom-H), 7.78–7.80 (t, H, Arom-H), 7.86–7.87 (d, 1H, Arom-H), 8.15–8.17 (d, 1H, Arom-H), 8.77 (brs, 1H, NH); Anal. Calcd. for C_25_H_24_N_4_O_4_S (476.55): C, 62.32; H, 4.79; N, 12.11; Found: C, 62.11; H, 5.14; N, 12.03.

##### 2-Methyl-3-[4-[[[(*n*-butylamino)carbonyl]aminosulfonyl]phenylethyl]-4(3H)-quinazolinone (VI-2-b)

White crystals (yield 64%); m.p. 163-165 °C; ^1^H NMR (DMSO-d_6_, 600 MHz) δ (ppm): 0.79–0.82 (t, 3H, CH_3_), 1.14–1.19 (m, 2H, CH_2_), 1.28–1.31 (m, 2H, CH_2_), 2.91–2.96 (q, 2H, CH_2_), 3.05–3.09 (t, t-like, 2H, CH_2_), 4.23–4.27 (t, 2H, N-CH_2_), 6.47 (t, 1H, CH_2_-NH), 7.47–7.52 (m, 2H, Arom-H), 7.57–7.59 (d, 1H, Arom-H), 7.77-7.81 (t, 2H, Arom-H), 7.84–7.86 (d, 2H, Arom-H), 8.11–8.13 (d, 1H, Arom-H), 10.4 (brs, 1H, NH); Anal. Calcd. for C_22_H_26_N_4_O_4_S (442.53): C, 59.71; H, 5.92; N, 12.66; Found: C, 59.19; H, 6.02; N, 12.05.

##### 2-Ethyl-3-[4-[[[(phenylamino)carbonyl]aminosulfonyl]phenylethyl]-4(3H)-quinazolinone (VI-3-a)

White crystals (yield 65%); m.p. 136-138 °C; ^1^H NMR (DMSO-d_6_, 600 MHz) δ (ppm1.22–1.28 (t, 3H, CH_3_), 2.78–2.82 (q, 2H, CH_2_), 3.05–3.08 (t-like, 2H, CH_2_), 4.24–4.29 (t, 2H, CH_2_), 6.95–6.98 (t, 2H, Arom-H), 7.26–7.29 (t, 2H, Arom-H), 7.45–7.47 (d, 2H, Arom-H), 7.48–7.51 (t, 2H, Arom-H), 7.54–7.55 (d, 1H, Arom-H), 7.61–7.63 (d, 1H, Arom-H), 7.78–7.81 (t, H, Arom-H), 7.86–7.87 (d, 1H, Arom-H), 8.13–8.14 (d, 1H, Arom-H), 8.64 (brs, 1H, NH), 12.0 (brs, 1H, NH); Anal. Calcd. for C_25_H_24_N_4_O_4_S (476.55): C, 63.01; H, 5.08; N,11.76; Found: C, 62.82; H, 5.60; N, 11.12.

##### 2-Ethyl-3-[4-[[[(cyclohexylamino)carbonyl]aminosulfonyl]phenylethyl]-4(3H)-quinazolinone (VI-3-b)

White crystals (yield 64%); m.p. 167–169 °C; ^1^H NMR (DMSO-d_6_, 600 MHz) δ (ppm): 1.0–1.15 (br, 6H, 3 CH_2_), 1.18–1.30 (br, 3H, CH_3_), 1.45–1.90 15 (br, 4H, 2 CH_2_), 2.76–2.82 (br, 2H, CH_2_), 3.0–3.12 (br, 2H, CH_2_), 3.25–3.35 (br, 1H, NH-CH), 4.26–4.32 (br, 2H, N-CH2), 6.95–6.98 (br, 1H, NH), 7.45–7.56 (br, 2H, Arom-H), 7.58–7.65 (d, 1H, Arom-H), 7.75–7.87 (br, 4H, Arom-H), 8.10–8.20 (br, 1H, Arom-H), 10.35 (brs, 1H, NH); Anal. Calcd. for C_25_H_30_N_4_O_4_S (482.60): C, 62.22; H, 6.27; N, 11.61; Found: C, 62.13; H, 5.98; N, 11.50.

##### 2-Ethyl-3-[4-[[[(*n*-butylamino)carbonyl]aminosulfonyl]phenylethyl]-4(3H)-quinazolinone (VI-3-c)

White crystals (yield 65%); m.p. 163–165 °C; ^1^H NMR (DMSO-d_6_, 600 MHz) δ (ppm): 0.80–0.83 (t, 3H, CH_3_), 1.15–1.20 (m, 2H, CH_2_), 1.27–1.32 (m, 2H, CH_2_), 2.78–2.83 (q, 2H, CH_2_), 2.93–2.95 (br, 2H, CH_2_), 3.04–3.08 (br, t-like, 2H, CH_2_), 4.25–4.26 (t like, 2H, N-CH_2_), 6.45 (brs, 1H, NH), 7.49–7.55 (m, 2H, Arom-H), 7.60-7.62 (d, 1H, j = 8 Hz, Arom-H), 7.77–7.80 (t, 2H, Arom-H), 7.85–7.87 (d, 2H, j = 7.5 Hz, Arom-H), 8.13–8.14 (d, 1H, j = 7.5 Hz, Arom-H), 10.52 (brs, 1H, NH); Anal. Calcd. for C_25_H_28_N_4_O_4_S (456.56): C, 60.51; H, 6.18; N, 12.27; Found: C, 60.28; H, 6.20; N, 11.73.

##### 2-*n*-Propyl-3-[4-[[[(phenylamino)carbonyl]aminosulfonyl]phenylethyl]-4(3H)-quinazolinone (VI-4-a)

White crystals (yield 71%); m.p. 136–138 °C; ^1^H NMR (DMSO-d_6_, 600 MHz) δ (ppm): 1.09–1.13 (t, 3H, CH_3_), 1.67–1.80 (m, 2H, CH_2_), 2.65–2.70 (t, 2H, CH_2_), 3.07–3.12 (t, 2H, CH_2_), 4.00–4.02 (q, 2H, CH_2_), 4.26–4.28 (t, 2H, CH_2_), 6.95–6.98 (t, 2H, Arom-H), 7.27–7.29 (t, 2H, Arom-H), 7.34 (s, 1H, NH), 7.43–7.53 (m, 4H, Arom-H), 7.58–7.63 (d, 1H, Arom-H), 7.76-7.79 (t, H, Arom-H), 7.83–7.85 (d, 1H, Arom-H), 8.13–8.15 (d, 1H, Arom-H), 8.64 (brs, 1H, NH), 12.0 (brs, 1H, NH); Anal. Calcd. for C_26_H_26_N_4_O_4_S (490.57): C, 63.99; H, 5.34; N, 11.42; Found: C, 63.43; H, 5.82; N, 11.52.

##### 2-*n*-Propyl-3-[4-[[[(cyclohexylamino)carbonyl]aminosulfonyl]phenylethyl]-4(3H)-quinazolinone (VI-4-b)

White crystals (yield 68%); m.p. 167–168 °C; ^1^H NMR (DMSO-d_6_, 600 MHz) δ (ppm): 1.07–1.13 (m, 6H, 3 CH_2_), 1.16–1.28 (m, 3H, CH_3_), 1.48–1.99 (m, 6H, 3 CH_2_), 2.66–2.70 (t, 2H, CH_2_), 3.04–3.07 (t, 2H, CH_2_), 3.25–3.35 (br, 1H, NH-CH), 4.25–4.28 (t, 2H, N-CH_2_), 5.58–5.60 (d, 1H, NH), 6.30–6.32 (br, 1H, NH), 7.45–7.51 (m, 2H, Arom-H), 7.58–7.60 (d, 1H, Arom-H), 7.77–7.83 (m, 4H, Arom-H), 8.12–8.14 (d, 1H, Arom-H); Anal. Calcd. for C_26_H_32_N_4_O_4_S (496.62): C, 62.88; H, 6.49; N, 11.28; Found: C, 63.30; H, 6.58; N, 10.38.

##### 2-*n*-Propyl-3-[4-[[[(*n*-butylamino)carbonyl]aminosulfonyl]phenylethyl]-4(3H)-quinazolinone (VI-4-c)

White crystals (yield 73%); m.p. 163–164 °C; ^1^H NMR (DMSO-d_6_, 600 MHz) δ (ppm): 0.78-0.85 (t, 3H, CH_3_), 0.95–1.05 (m, 3H, CH_3_), 1.11–1.21 (m, 2H, CH_2_), 1.26–1.33 (m, 2H, CH_2_), 1.68–1.75 (m, 2H, CH_2_), 2.64–2.70 (t, 2H, CH_2_), 2.91–2.96 (q, 2H, CH_2_), 3.04–3.08 (t, 2H, CH_2_), 4.24–4.28 (t, 2H, N-CH_2_), 6.45–6.46 (t, 1H, NH), 7.47-7.53 (m, 2H, Arom-H), 7.58–7.60 (d, 1H, Arom-H), 7.77–7.80 (t, 2H, Arom-H), 7.83–7.85 (d, 2H, Arom-H), 8.11–8.13 (d, 1H, Arom-H), 10.5 (brs, 1H, NH); Anal. Calcd. for C_24_H_30_N_4_O_4_S (470.58): C, 61.26; H, 6.43; N, 11.91; Found: C, 61.37; H, 6.88; N, 11.74.

##### 2-*iso*-*pr*-3-[4-[[[(Cyclohexylamino)carbonyl]aminosulfonyl]phenylethyl]-4(3H)-quinazolinone (VI-5-a)

White crystals (yield 68%); m.p. 167–168 °C; ^1^H NMR (DMSO-d_6_, 600 MHz) δ (ppm): 0.85–0.98 (m, 6H, 3 CH_2_), 1.02–1.04 (d, 6H, 2 CH_3_), 1.30–1.33 (m, 3H, CH), 1.40–1.47 (br, H, CH_2_), 1.51–1.54 (br, 2H, CH), 2.88–2.94 (t, 2H, CH_2_), 3.09-3.16 (m, 1H, CH_2_), 4.17–4.20 (t, 2H, N-CH_2_), 6.50–6.52 (d, 1H, NH), 7.29–7.31 (d, 2H, Arom-H), 7.36-7.40 (t, 1H, Arom-H), 7.60–7.69 (m, 4H, Arom-H), 7.98–8.00 (d, 1H, Arom-H), 10.4 (Brs, 1H, NH); Anal. Calcd. for C_26_H_32_N_4_O_4_S (496.62): C, 62.88; H, 6.49; N, 11.28; Found: C, 62.55; H, 6.22; N, 10.95.

##### 2-*iso*-Propyl-3-[4-[[[(*n*-butylamino)carbonyl]aminosulfonyl]phenylethyl]-4(3H)-quinazolinone (VI-5-b)

White crystals (yield 73%); m.p. 163–164 °C; ^1^H NMR (DMSO-d_6_, 600 MHz) δ (ppm): 0.79–0.83 (t, 3H, CH_3_), 1.11–1.13 (d, 6H, 2CH_3_), 1.16–1.20 (m, 2H, CH_2_), 1.28–1.32 (m, 2H, CH_2_), 2.92–2.94 (m, 1H, CH_2_), 3.05–3.09 (t, 2H, CH_2_), 4.31–4.35 (t, 2H, N-CH_2_), 6.44 (t, 1H, NH), 7.45–7.50 (m, 3H, Arom-H), 7.58–7.60 (d, 1H, Arom-H), 7.78–7.83 (m, 3H, Arom-H), 8.13–8.15 (d, 1H, Arom-H), 10.5 (brs, 1H, NH); Anal. Calcd. for C_24_H_30_N_4_O_4_S (470.58): C, 61.26; H, 6.43; N, 11.91; Found: C, 61.19; H, 6.18; N, 11.52.

##### 2-Benzyl-3-[4-[[[(phenylamino)carbonyl]aminosulfonyl]phenylethyl]-4(3H)-quinazolinone (VI-6-a)

White crystals (yield 71%); m.p. 136–138 °C; ^1^H NMR (DMSO-d_6_, 600 MHz) δ (ppm): 2.94 (br, 2H, CH_2_), 3.44 (br, 1H, CH), 4.01 (brs, 1H, CH), 4.12 (br, 2H, CH_2_), 6.96–6.98 (br, 2H, NH + Arom-H), 7.05 (br, 1H, Arom-H), 7.27 (br, 2H, Arom-H), 7.34 (brs, 1H, Arom-H), 7.40 (br, 1H, Arom-H), 7.47–7.58 (br, 6H, Arom-H), 7.66–7.71 (br, 3H, Arom-H), 7.84 (br, 1H, Arom-H), 8.23 (br, 1H, Arom-H), 8.67 (br, 1H, Arom-H), 11.6–11.8 (br, 1H, NH); Anal. Calcd. for C_30_H_26_N_4_O_4_S (538.62): C, 66.90; H, 4.87; N, 10.40; Found: C, 66.92; H, 5.07; N, 10.32.

##### 2-Benzyl-3-[4-[[[(cyclohexylamino)carbonyl]aminosulfonyl]phenylethyl]-4(3H)-quinazolinone (VI-6-b)

White crystals (yield 68%); m.p. 167–168 °C; ^1^H NMR (DMSO-d_6_, 600 MHz) δ (ppm): 0.77–0.80 (t, 3H, CH_3_), 1.12–1.20 (m, 2H, CH_2_), 1.22–1.35 (m, 2H, CH_2_), 2.75–2.79 (t, 2H, CH_2_), 2.90–2.95 (m, 2H, CH_2_), 4.15–4.18 (t, 2H, CH_2_), 4.19 (s, 2H, CH_2_), 6.45 (br, 1H, NH), 7.27–7.29 (d-like, 4H, Arom-H), 7.34–7.38 (m, 3H, Arom-H), 7.52–7.58 (t, 1H, Arom-H), 7.65–7.68 (d, 1H, Arom-H), 7.81–7.84 (d, 3H, Arom-H), 8.16–8.18 (d, 1H, Arom-H), 10.6 (br, 1H, NH).; Anal. Calcd. for C_30_H_32_N_4_O_4_S (544.66): C, 66.15; H, 5.92; N, 10.29; Found: C, 66.46; H, 5.96; N, 10.56.

#### 3.1.6. 2-Phenyl-3-[4-[[[(cyclohexylamino)carbonyl]aminosulfonyl]phenyl]-4(3H)-quinazolinone (VII)

A mixture of (V) (0.05 mole) and anhydrous K_2_CO_3_ (0.1 mol, 13.8 g) in dry acetone (100 mL) was stirred and refluxed for 2 h. At this temperature, a solution of hexyl isocyanate (0.075 mole) in dry acetone (20 mL) was added in a dropwise manner. The reaction mixture was further refluxed overnight. The solvent was then removed under vacuum and the solid residue was dissolved in water and then acidified with a sufficient amount of 2N HCl. The separated product was recrystallized from ethanol (VII).

White crystals (yield 71%); m.p. 224–226 °C; ^1^H NMR (DMSO-d_6_, 600 MHz) δ 1.10–1.14 (m, 4H, CH, Cyclohexyl protons), 1.12–1.26 (m, 2H, Cyclohexyl protons), 1.45–1.55 (br, CH, Cyclohexyl protons), 1.60–1.67 (m, 4H, Cyclohexyl protons), 3.29–3.30 (br, 1H, N-CH), 6.34–6.35 (d, 1H, NH), 7.23–7.26 (m, 3H, Arom-H), 7.38-7.39 (d, 2H, Arom-H), 7.59–7.64 (m, 3H, Arom-H), 7.79–7.84 (d, 2H, Arom-H), 7.91-7.94 (t, 1H, Arom-H), 8.21–8.22 (d, 1H, Arom-H), 10.3–10.5 (br, 1H, NH); Anal. Calcd. for C_27_H_26_N_4_O_4_S (502.58): C, 63.65; H, 4.01, 11.13; Found: C, 63.32; H, 3.86; N, 10.94.

### 3.2. Antidiabetic Activity

Adult male albino Wister rats of 8–10 weeks old with a body weight of 150–200 g were used in this investigation. One hundred and fifteen rats were arranged in twenty-three groups (n = 5). Rats of the first group were normal, the other were made diabetic and were divided into twenty-two groups. Diabetes was induced by IP injection of streptozotocin (STZ) in a dose of 55 mg/kg. One diabetic group received CMC as a solvent and served as control. The other groups of diabetic rats were treated with glibenclamide as a reference standard and the tested compounds from III-1 to VII. Glibenclamide and the tested compounds were suspended in CMC (5%) and were given to rats orally by gavage. Rats were treated with glibenclamide and the test compounds in a dose of 2 m/kg for 6 days, followed by 6 days withdrawal. Blood samples were withdrawn from the tail tip and blood glucose levels were determined two hours later on the six days of the dose, on the sex day after discontinuation of the treatment. Statistical calculation was carried out using the paired “*t*” test. The mean values were compared, and the values were considered significant at *p* < 0.05 [43].

### 3.3. Molecular Modeling

#### 3.3.1. Docking Studies

The crystallographic structure of PPARγ was retrieved from Protein Data Bank [PDB ID- 1FM6, resolution 2.1 Å] (http://www.pdb.org, accessed on 10 May 2019), and considered as a target for docking simulations. The docking analysis was performed using MOE [40] software to evaluate the free energies and binding mode of the designed molecules against PPARγ. At first, the crystal structure of PPARγ was prepared by removing water molecules and retaining only one chain and its co-crystallized ligand, rosiglitazone. Then, the protein structure was protonated, and the hydrogen atoms were hidden. Next, the energy was minimized, and the binding pocket of the protein was defined. The 2D structures of the synthesized compounds, rosiglitazone and glibenclamide were sketched using ChemBioDraw Ultra 14.0 and saved as MDL-SD format. Then, the saved file was opened using MOE and 3D structures were protonated. Next, energy minimization was applied. Before docking the synthesized compounds, validation of the docking protocol was carried out by running the simulation only using the co-crystallized ligand and low RMSD between docked and crystal conformations. The molecular docking of the synthesized compounds and reference ligands (rosiglitazone and glibenclamide) were performed using a default protocol. In each case, ten docked structures were generated using genetic algorithm searches.

#### 3.3.2. ADMET Studies

Lipophilicity, solubility, drug likeness, and GIT absorption were carried out using SwissADME software [41] while toxicity study was carried out using T.E.S.T software [42]. These programs are free software.

## 4. Conclusions

This study showed synthesis of a novel series of quinazoline-sulfonylurea derivatives as modified structures of glibenclamide. The synthesized compounds were exposed to two biological tests using in-vivo antihyperglycemic activity against STZ induced hyperglycemic rats using glibenclamide as a reference drug. The first experiment was performed to measure antidiabetic activity of the tested compounds using doses of 2 mg/kg. The second experiment was performed to examine the prolonged antidiabetic effect of the tested compounds. Compounds VI-6-a, V, IV-4, VI-4-c, IV-6, VI-2-a, IV-1, and IV-2 had better antidiabetic activity than the reference glibenclamide. Compounds IV-1, VI-2-a, IV-2, V, and IV-6 had more prolonged antidiabetic activity than glibenclamide. Molecular modeling study showed good binding affinities for the tested compounds with PPARϒ receptors. Additionally, in silico ADMETT studies showed promising pharmacokinetic parameters for the tested compounds.

## Data Availability

Not applicable.

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
