# Peer review of "Design, Synthesis, Molecular Modeling and Anti-Hyperglycemic Evaluation of Quinazoline-Sulfonylurea Hybrids as Peroxisome Proliferator-Activated Receptor Gamma (PPARγ) and Sulfonylurea Receptor (SUR) Agonists"

_ijms, 2022, doi:10.3390/ijms23179605_

Round 1

Reviewer 1 Report

Introduction of quinazoline and hybridization with rosiglitazone structure has certainly resulted in compounds with greater efficacy.

please find the attached document with highlighted words and sentences with comments. Manuscript will greatly benefit with rewriting and presentation of the data in a concise fashion. 

Also, work from ref 35-37 should be presented along with structures so that progression of work can be followed and the novelty of current work can easily be established.  

Many tables and figures are not needed and can be removed. 

Author Response

Dear Reviewer,

Thank you for sending us that useful comments and suggestions on our manuscript. We have modified the manuscript accordingly, and detailed corrections and comments are listed below point by point:

1) Highlighted words and sentences to be corrected.

All highlighted words and sentences were corrected or modified throughout the whole manuscript as required.

2) “Auxochromacophore” what does this mean and why is it relevant here.

Auxochrome is a group of atoms which will impart a particular color when attached to a chromophore (part of the molecule responsible for color appearance). It is included to explain a type of antidiabetic quinazoline with different characters. The sentence contains this word was deleted.

3) The study was performed in animals, so it is invivo not invitro.

The word was corrected.

4) Why is Gibenclamide treated rats have higher glucose after treatment.

We apologize for this mistake. There was a missing number 336. It is missed by mistake, but it is present in the second table. The correct glucose level before treatment was 336±23.

5) Table 1 not needed, table 2 is enough.

Table 1 was removed

5) Why are these values same as 6 days.

Kindly be noted that it is one experiment, but it is done in two steps. First step was measuring the hypoglycemic effect of tested compounds in a dose of 2mg/kg for 6 days. Second step was 6 days withdrawal or discontinuation then measuring the blood glucose level, so the values are the same. Table 1 was removed as required while table 2 can explain all the results.

5) Label Y axis.

Y axis was labeled in the two graphs.

6) Work from ref 35-37 should be presented along with structures

Structures were added in figure 3 with some explanation.

Sincerely.

Reviewer 2 Report

Many formal and grammatical errors (see enclosed pdf)

Author Response

Dear Reviewer,

Thank you for sending us that useful formal and grammatical  corrections for our manuscript. We have modified the manuscript accordingly, and detailed corrections and comments are listed below point by point.

Explain abbreviations.

Abbreviations were explained.

Many formal and grammatical errors.

All errors were corrected throughout the whole manuscript. The corrected words and statements were highlighted in yellow color.

Glibenclamide is not included in table 4.

The pharmacokinetic values of Glibenclamide were included in table 4.

Sincerely.

Round 2

Reviewer 1 Report

Figure 2 still has incorrect structures 

Author Response

Dear: Reviewer

We would like to thank you for your revision of our manuscript. Structures in figure 2 were corrected.

Thank you.